# Qualitative Analysis of Micro-System-Level Factors Determining Sport Persistence

**DOI:** 10.3390/jfmk9040196

**Published:** 2024-10-18

**Authors:** Bence Tamás Selejó Joó, Hanna Czipa, Regina Bódi, Zsuzsa Lupócz, Rozália Paronai, Benedek Tibor Tóth, Hanna Léna Tóth, Oszkár Csaba Kocsner, Buda Lovas, Csanád Lukácsi, Mátyás Kovács, Karolina Eszter Kovács

**Affiliations:** 1Institute of Psychology, University of Debrecen, 4032 Debrecen, Hungary; benui1999@gmail.com (B.T.S.J.); czipahanna@gmail.com (H.C.); bodiregina99@gmail.com (R.B.); zsuzsalupocz@gmail.com (Z.L.); prozka2002@gmail.com (R.P.); bene.toth@gmail.com (B.T.T.); hannalenatoth@gmail.com (H.L.T.); kocsner.oszi@gmail.com (O.C.K.); lukacsi.csanad.pal@gmail.com (C.L.); 2Institute of Psychology, Eötvös Loránd University, 1064 Budapest, Hungary; lovasbuda@gmail.com; 3Faculty of Chemical Technology and Biotechnology, Budapest University of Technology and Economics, 1111 Budapest, Hungary; matyikaa2000@gmail.com

**Keywords:** sport persistence, ecological model, qualitative research, individual motivation, grounded theory

## Abstract

**Background/Objectives**: Sport persistence is the embodiment of sports performance and mental toughness. It refers to our attempts concerning the performance plateau, failures, injuries, or even the resolution and processing of stressful situations associated with success and positive events. In our research, we used qualitative methods based on Bronfenbrenner’s socio-ecological model to investigate the factors influencing sport persistence among high school and university athletes. **Methods**: The research was based on semi-structured interviews with 133 athletes. ATLAS.ti software and the grounded theory methodology were applied for data analysis. Our analysis grouped the responses according to Bronfenbrenner’s categorisation system, highlighting motivational factors at the microsystem level. Our research question was as follows: What kind of factors dominate the development of sport persistence among adolescent (high school) and young adult (university) athletes along Bronfenbrenner’s dimension of the microsystem? **Results**: Regarding the microsystem, family, peers, and coaches were mentioned as influential factors. Concerning the family, general, person-specific, family value-related, future-oriented, introjected, and disadvantage-compensating motivational components were identified. General, individual, community and relational factors were identified concerning peers. Concerning the coach, general, individual, community, and coach personality-driven motivational segments were detected. **Conclusions**: By recognising the complex interplay of systemic factors, we can design interventions targeting these factors at various socio-ecological levels, promoting youth sports and increasing physical activity among young people. These findings instil hope and motivation for the future of sports and physical activity.

## 1. Introduction

The concept of sport persistence, i.e., behavioural commitment to sport, involves a combination of sports performance and engagement, resulting in an athlete’s unwavering dedication to their chosen sport. This incorporates cultivating attributes such as adaptive coping mechanisms, positive personality traits, and resilience [1]. The athlete is, therefore, not merely persistent in the sporting activity but also qualitatively committed to it. This refers to the attempts to resolve, process, and utilise stressful circumstances associated with performance plateaus, failures, injuries, successes, and positive events. The concept of sport persistence encompasses this behaviour and performance. It is an under-researched area in international practice, as research in this field typically focuses on sporting habits, sports motivation, and engagement. A shortcoming of previous research is that the study of the manifestation of sport persistence often focuses on a single component, leaving aside the complex, multi-psychological and non-psychological, individual and social factors. In addition, researchers often treat sport persistence as a dichotomous variable: whether the athlete is still pursuing sport or has dropped out. However, like motivation, it would be useful to assess sport persistence as a continuous variable along a spectrum. Overall, sport persistence can be regarded as a performance indicator that refers to a person’s performance through sustained physical activity (regardless of the level of activity).

Young athletes, particularly the talented ones, may drop out of sport due to individual psychological, social, and contextual factors during their school years before reaching their peak performance [2,3]. A combination of gender, socio-economic status, and peer (parents, coaches, peers) support factors may predict sport persistence and its opposite, dropout from the sport in childhood [4,5]. Therefore, both sport persistence and dropout can be identified as multifactorial and complex phenomena, strongly influenced by different socio-cultural backgrounds and behavioural factors, as well as personal characteristics, types of sport, attitudes, and motivations [6].

We use Bronfenbrenner’s [7] socio-ecological model as a starting point to explain the differences in the manifestation and background of sport persistence. The fundamental precondition of Bronfenbrenner’s socio-ecological model is that individuals are intimately connected to and influenced by their environment. In particular, Bronfenbrenner argues that individual behaviour can be understood in terms of four environmental systems: micro-, meso-, exo-, and macrosystems. Like a Matryoshka doll set, these different systems are layers of nested systems, with the innermost layer representing the self. The microsystem comprises a complex of close relationships, such as close neighbours, family members, peers, school class, and workplace environment. The mesosystem represents the second layer. It is the context in which the microsystem components mentioned above are inter-related. The mesosystem, thus, refers to the relationships between microsystems. The exosystem is the third layer and refers to supporting environments where individuals are inactive participants. Exosystem factors that influence sport participation include formal settings and the physical characteristics of the sports environment, such as community centres, parks, recreation centres, sports clubs, and sports facilities. The fourth and outermost layer of Bronfenbrenner’s model is the macrosystem, which represents an overarching consistency of the previously defined systems (micro-, meso- and exosystem) at the level of society as a whole.

According to Bronfenbrenner’s [7] ecological model (see Figure 1), the following factors are essential for the development and persistence of the athlete:Individual level: age, gender, gender orientation, physical characteristics (weight, height, BMI), physical health (illnesses), mental health (health awareness and behaviour, risk-taking behaviour, anxiety, coping, resilience, well-being, religious/spiritual well-being, sense of coherence, self-efficacy, future orientation, burnout, loneliness), other activities (educational career, employment);Microsystem: family characteristics (family structure, number of siblings, sibling order, parental employment, socio-economic status and well-being, family attachment and relationships, family social relations), friend characteristics (strength, type, quality, nature of), sports environment (relationship with sports partners, relationship with coaches, training characteristics, relationship with clubs and institutional climate);Mesosystem: family–school relationship, peer–parent relationship, family–school–peer triangle;Macrosystem: social trends, political trends and regulations, cultural characteristics and values.

**Figure 1 jfmk-09-00196-f001:**
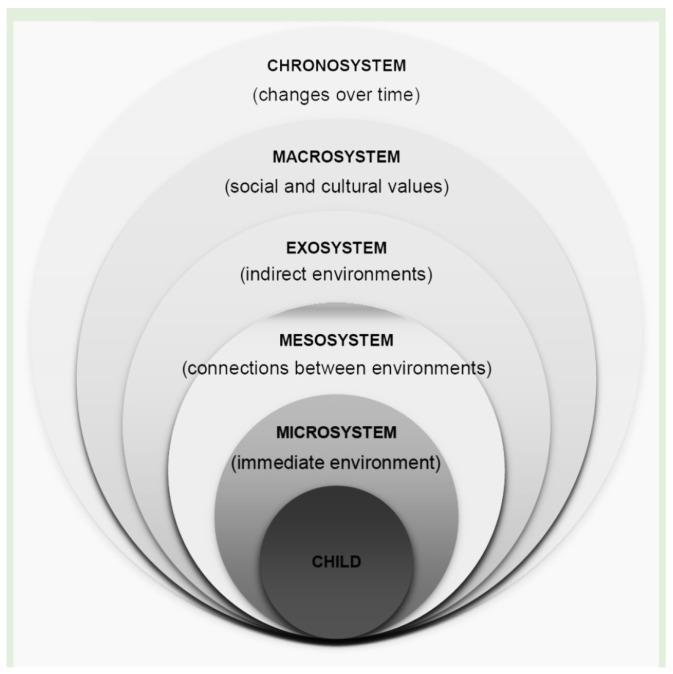
Bronfenbrenner’s ecological model [7].

Bauman et al.’s [8] model uses a multi-level comprehensive framework that categorises all the variables influencing physical activity into levels (see Figure 2). Because of its ecological nature, the model encompasses the interactions between individuals and their social and physical environments. The fundamental premise is that a comprehensive understanding of the determinants of physical activity provides the opportunity to develop multi-level action plans that are significantly more likely to succeed. The model incorporates individual variables, such as biological and psychological factors, and allows for a broad examination of micro, meso, exo, and macro variables. The combination and interaction of political, environmental, and global factors are considered to have a wide range of impacts on sporting activity, but they have only been tangentially examined so far.

Following the models of Bronfenbrenner and Bauman, in general, it is rather uncommon to highlight sociodemographic and career-related background variables as elements that contribute to sport persistence, although it is well known that a combination of socio-economic, cultural, and parental support factors most effectively predicts childhood sport participation. Numerous studies indicate the considerable influence of gender on sport persistence, often demonstrating that male athletes exhibit greater persistence, a finding that aligns with earlier research on long-term participation in sports [5,9,10]. Additionally, assessing the effects of career-related variables is crucial, particularly concerning elite sport participation and persistence. For elite athletes, sports biography and experience hold significant importance. Concerning individual psychological factors, in our systematic literature analysis focusing on the existing research on sport persistence, we detected nine groups of variables, named personality, motivation, orientation, learning and development, commitment, positive feelings, negative feelings, future benefits, and health. These groups and the variables belonging to them are the most typical factors investigated in papers focusing on sport participation, dropout, and persistence [11]. Persistence is seen as a positive variable, exhibiting greater advantages as its level increases, provided it remains within an optimal range. Consequently, it demonstrates a positive correlation with other favourable variables, such as coping flexibility, optimism, self-assessment, and self-image, while showing a negative correlation with adverse variables like anxiety, stress, pessimism, and narcissism. This relationship holds true not only in terms of correlation but also in the direction of influence; positive variables exert a supportive effect, whereas negative variables impose a hindering effect on persistence in sports [12,13]. Concerning the microsystem, following Bronfenbrenner’s categorisation, the relevance of the family, peers, and coaches should be emphasised, each having a supporting impact on persistent sporting behaviour [3,14,15]. Measurement of the mesosystem-level factors like the family–school relationship and family–coach relationship is under-represented in this area, although some attempts could be found. For instance, previous investigations showed that the integration of school and sport can have a direct impact on sport persistence [16]. Lastly, concerning macrosystem factors, subjective norms, cultural expectations, and beliefs may have a notable impact on sport persistence [17]. A positive sport culture that values effort, sportspersonship, and personal development can foster a long-term commitment to the sport. On the other hand, a negative or toxic sport culture can discourage athletes and lead to disengagement.

The present research focuses on the microsystem-level factors underlying sport persistence. In this context, we have formulated the following research question: *what are the dominant factors that shape the development of sport persistence among adolescent (high school) and young adult (university) athletes, as viewed through the microsystems framework proposed by Bronfenbrenner, regarding the family, peers, and coach?* In order to address the research question, the participants’ responses were grouped according to content categories and compared with previous findings in the literature.

## 2. Presentation of the Research

### 2.1. Sample

The data were collected through theoretical sampling. In the initial stage of the study, a wider range of potential interviewees was identified based on the criteria set by experts in the field. As the analysis of the interviews progressed, additional subjects were selected from this range based on the emergence of new concepts. The sampling was guided by clarifying the categories identified through open, axial, and selective coding. The expert criteria defining the broader range of potential interviewees were as follows: (1) participation in sport (any individual or team sport, as appropriate) with some regularity (at least three times a week); (2) pursuit of secondary or tertiary education; (3) age between 14 and 25 years. The research involved Hungarian participants, living in the North Great Plain region (see Table 1). A total of 133 high school and university-level athletes were surveyed qualitatively using a semi-structured interview technique. The recruitment process entailed contacting sports clubs, students enrolled in sports schools, and university students engaged in athletic pursuits.

### 2.2. Methods

The research used a semi-structured interview methodology for data collection. The interview schedule included four blocks, including sociodemographic background, sport-specific questions, and competitive athletic and sport persistence issues along the levels of the ecological model; this block could be divided into five sub-sections (see Table 2).

The data were analysed using the grounded theory (GT) method, which is inductive and iterative. Grounded theory [18,19] was employed to move beyond the confines of our existing knowledge on the subject and to develop theoretical justifications based on the data obtained from the interview analyses while progressing through the stages of abstraction. The semi-structured interview questions served as the starting point for the inquiry, allowing for free-associative reflection to cultivate a discussion about matters the interviewees might deem necessary related to the subject matter [20,21]. Concurrently, deductive category analysis was employed to structure the analysis. A qualitative, category-driven text interpretation was employed, emphasising the potential for feedback and intersubjective testing. The content analysis was conducted using ATLAS.ti software (22.2.5 Student version). In accordance with the constructivist approach to meaning-making espoused by GT, our analysis focuses on the ‘what’ and ‘how’ questions, with the aim of identifying the specific components that shape the sport persistence of young people engaged in sporting activities at the high school and university level. The GT techniques were employed to identify themes and patterns and create categories of factors involved in the formation of sport persistence.

The type of qualitative study and the use of grounded theory indicates moving in flexible environments, in terms of facing unpredictable fields of action. For this reason, the research was structured in six phases, including the following tasks carried out by the research group members:Phase 1: definition of the sample (the head of the research group, based on the research project and its plan, detected the potential sample);Phase 2: validation of the interview script (all members were asked to think through the potential interview questions, then a discussion was carried out in a research group meeting);Phase 3: application of the interviews (all members were involved in the interview process, including recruitment and application as well);Phase 4: transcription and analysis (five research group members [HC, RP, HLT, OCsK, CsL] were involved in the preparation of the transcripts and seven in the analysis [BTSJ, RB, ZsL, BTT, BL, MK, KEK]);Phase 5: coding and categorisation (Seven research group members [BTSJ, RB, ZsL, BTT, BL, MK, KEK] were involved in the creation of codes. In the initial stage, the analysis involved the identification and coding of incidents. Afterwards, the preliminary codes were compared with other codes, and the codes were grouped into categories. The incidents within each category were then compared with those in other categories. Then, the future codes and categories were compared. Subsequently, the new data were subjected to comparison with the information that had been collected at an earlier stage of the analysis.);Phase 6: interpretation (all members were asked to support the interpretation of the results).

The iterative process entailed the application of both inductive and deductive reasoning. The interview analyses employed a combination of inductive, deductive, and abductive reasoning. Grounded theory establishes the importance of highlighting contextual influences, e.g., intrapersonal characteristics (ontosystem), interpersonal relationships (microsystem), community influences (mesosystem), and cultural values and national characteristics (macrosystem), which can be used to reveal socio-psychological influences [19]. Taking this into account, we explored and thematically sequenced the characteristics of micro and macro environmental influences in the interview transcripts. The content analysis was conducted using ATLAS.ti software (22.2.5 Student version), however, the programme was only used to synthesise coding. In the analysis, the data were grouped around microsystem-level factors, which were developed by combining information from the interviews with concepts from the literature. The ATLAS.ti software proved to be a particularly useful tool for managing qualitative data and exploring the relationships between complex concepts. ATLAS.ti provided the possibility to precisely manage the coding process and to organise themes and sub-themes, thus facilitating the creation of typologically ordered categories. The coding phases were divided into four phases, following the constructivist line of grounded theory (GTC) [22]. In the first phase, we familiarised ourselves with the interview texts through multiple readings and listening, followed by semantic-based open coding carried out in the second phase. In the third stage, based on the samples of codes, we categorised them into categories [23]. In the fourth stage, we repeated the analysis cycles to examine and refine the code classifications of the categories. In the analysis, we followed the principle of theoretical and personal triangulation, whereby we examined the same research question and the phenomenon of the family–school relationship using different theoretical frameworks and perspectives. This approach allowed us to explore different aspects of the phenomenon by comparing different theories and to build a more comprehensive picture based on the different perspectives (Flick [24]). Therefore, the present study resorted to the quality criteria exposed by Flick [24], known as credibility, data-rich elicitation, commitment to fieldwork, and confirmability.

## 3. Results

### 3.1. Family-Related Impact

Figure 3 shows the main and sub-categories created concerning the family-related factors influencing sport persistence. In the content analysis, six motivational segments were identified: general (non-concrete), personal, family values, future-oriented, introjected, and disadvantage compensation. For the general, non-specific motivation, interviewees did not specify the nature of the support, but they certainly reported a supportive effect and attitude. This motivation segment could not be further subdivided into subgroups (see Table 3).

In the case of compensating for disadvantage, there were responses where sport is used as a means of escaping from some negative circumstance or context. The segment comprises two subgroups. The first is the promotion of excellence and talent, which emerged among young athletes growing up in particularly disadvantaged circumstances, suggesting that sport can be a means of social inclusion and successful coping. And detachment from the family suggests that sport provides an opportunity to break away and detach from the family, which can support the young athlete’s aspirations for independence.

In the case of introjected motivation, responses emerged that were motivated by an external source, but were also embedded in the individual’s internal values. Intrinsic expectation refers to the interviewees’ need to maximise their sporting performance to the best of their ability and to meet their own expectations. The desire to prove themselves refers to the subjects’ strong desire to prove themselves to their families and to gain the recognition of their relatives through their sporting performance, which gives them satisfaction and motivation to persevere in sport.

In terms of the person-related support, family motivation and sporting history were prominent in the interviewees’ responses. The subjects’ commitment to sport is strongly influenced by family background and sporting history. Many of them had parents or family members who played sport, and this motivated them to participate in sport. Some participate in sport to make their parents proud, others because sport has always been in their family and has been part of their everyday life since childhood. The motivation of family members also has a significant influence on the development of sporting persistence. Some subjects are motivated by their parents, such as a mother or father who is a PE teacher. Others play sport to motivate their family members, to encourage them to adopt a healthy lifestyle, and to find a common hobby. In addition, perseverance plays an important role, as an example they want to set for their families, even in the event of injury.

In the case of support for the future, future family motivation emerged, which refers to the interviewee’s conscious preparation of the values and behaviours that he or she would like to pass on to his or her children in the future. The motivation for future livelihood suggests that sport could be a career goal that could serve as a means to financial well-being. They would like to earn a living from sport, which will allow them to earn money to spend on their families and provide them with opportunities in the future. In the case of motivation for development, it is clear that subjects are highly motivated by the support and example of their family. Parents provide all the support they need to achieve their goals and progress. They are an inspiring influence as they are active in sports and work hard. Finally, outperformance also emerged as a motivation, expressing the desire for the athlete to achieve a more successful sporting career and even better performance than their parents.

Finally, in the case of the support related to family values, health maintenance could be distinguished, suggesting that the subjects’ motivation to exercise is primarily to maintain health and good physical condition. Many people exercise to avoid diseases that affect their relatives and to stay fit and healthy. They also want to avoid being overweight and live as long as possible in good health with their loved ones. The shared values segment refers to the role of family values and education. In this case, interviewees emphasised the role of family values in regular sporting activity and also stressed the importance of regular exercise, a need they had been keen to develop since childhood. For the joint activity component, sport is a shared activity and programme for the subjects with their family members, giving them the opportunity to be together, talk, and support each other. The interest and support of family members plays an important role in their engagement in sport. In the case of the source of pleasure motivation factor, sport is a source of happiness, which also brings pleasure to their family. They want to make them proud and enjoy the support and happiness of their family members when they see their success and happiness. This family joy and support is a strong motivation for them. Relaxation as a drive suggests that sport is an important relaxation tool. It helps relieve tension and provides relaxation and pleasure, which contributes to their physical and mental well-being. Sports enthusiasm means that for the interviewees, sports enthusiasm is deeply embedded in their lives. They love sport, which gives them pleasure and freedom. Their passion for sport also provides support and motivation as they follow the example of their family members and coaches. They also value the social and therapeutic role of sport. Finally, the motivation of mutual support was isolated, suggesting that athletes can motivate their family members and, in return, receive encouragement and support from them in all forms, which includes attending training sessions and competitions. They can also support their families through their success in sport.

### 3.2. Peer Influence

Figure 4 shows the main and sub-categories created concerning the peer-related factors influencing sport persistence. There was also a dominance of factors supporting persistence among peers. A certain group of factors could be registered in this case as well, which represented a kind of general non-concretised motivation and support. In addition, three larger groups of motivational factors could be established (see Table 4).

One major group of motivation is community motivation, which typically includes outward, peer-oriented persistence components. Collective self-consciousness is the cultivation and maintenance of team unity through the cultivation and maintenance of friendships and the manifestation of team spirit, which approaches development from a collective rather than an individual perspective. Altruism is a similar component, but in this case, it was explicitly emphasised by respondents that under no circumstances would anyone be left alone, and that the success of the team was more important than individual performance. Joint activity reflects the actions, time spent together, and the pleasure this brings. The peer motivation component is specifically aimed at motivating sporting partners to improve and retain peer performance in sporting activities and to sustain sport-related joint activities. Peer comparison implies a desire to compete, either with sporting peers or with non-sports peers, but also carries with it a need for peer acceptance. The social support provided by sporting peers reflects a socially supportive anticipation that incorporates love of the sporting partner, camaraderie, good team structure and listening to each other. Belonging somewhere emphasises the role of belonging, the team as a family, and the cohesive community, which is also significant for sport persistence. Finally, the positive environment refers specifically to the environment and climate, including the retaining and motivating power of people, positive feedback, and attitudes.

In the case of individual motivation, some kind of peer-related yet individual driving force emerged. The desire to prove oneself refers to a person’s desire to show peers the abilities, skills and knowledge they possess, which they would like to prove to their peers. In the case of individual development, the focus is on how peers can contribute to the athlete’s personal development—to the maximum performance that can help them reach peak performance and bring out the best in themselves. Health promotion refers to the maintenance of the physical and mental health of the individual, in which the role of peers can be significant, as their behaviour can also influence the health behaviour of the athlete in a positive direction. The recognition component involves positive feedback from the environment, in particular praise and appreciation, and a sense of pride in the athlete from peers.

Finally, relational motivation could be differentiated as a segment that supports sport persistence. This included networking, which emphasised the roles of teamwork and teambuilding in particular, which may serve as resources that could have a role in maintaining sport persistence. Setting an example includes a need for peers, which allows the athletes to think of themselves as role models who can set a positive example for their immediate and broader surroundings.

### 3.3. The Impact of the Coach

Figure 5 shows the main and sub-categories created concerning the coach-related factors influencing sport persistence. A small number of interviewees said that the coach’s role was irrelevant to them, as they play a type of sport or practice a level of sport that does not require a coach to train them (see results in Table 5). There was a separate category of a ‘not influenced’ type of responses, where the coach’s role is not emphasised for individual or sport-specific reasons (as in the case of the ‘not relevant’ type responses). However, the vast majority of responses refer to the positive, supportive role of the coach. The types of support can be grouped into several categories, further subdivided into subgroups. General support is presented as general support, as well as total support. For the former, respondents refer to a kind of general, generic support, while for the latter, they refer to all-round support.

Community motivation emerged as a separate group, which includes influencing factors that focus on the coach, peers, and the community. Altruism refers to team spirit, which refers to helping and supporting each other. The team above all else attitude and the emphasis on the cardinal role of the coach as team leader fall into this group. The other sub-category belonging to this larger motivational group is belonging to something that specifically reflects a sense of love, attachment, and belonging to a community. Related are statements that emphasise the importance of contribution and prioritise the interests of the team over those of the individual.

In the case of individual motivation, we observe the clear presence of an individual intrinsic motivational component. In the case of the internal expectation component, there is an internalised expectation of the coach’s role in talent development and the associated coach care, which may result in coach recognition. However, the focus is on the coach, not the athlete. In the case of the desire to prove component, there is a manifestation of the individual’s internal motivation to prove to and for the coach the athlete’s sporting activities and achievements. This includes individual development, perseverance and commitment, achievement, and achievement. Recognition is a type of extrinsic motivation segment that is well related to the previous components. However, it is explicitly indicated that the athlete wants to earn the coach’s recognition, as the athlete sees professional knowledge in them. Positive feedback and recognition from the coach as a reference person is essential for the athlete’s career and career development. Health promotion as a drive is a separate category based on the coach’s pattern of health promotion behaviour. The athlete internalises this and it becomes a kind of internal need for a healthy and health-conscious lifestyle. The motivational component of development involves acquiring and practically applying information and knowledge material from the coach. In parallel, some mentioned the coach’s love of the sport and psychological qualities outside professionalism, such as the contribution to becoming a better person.

Finally, motivation by the coach’s person could be separated into a larger group, including several subgroups. The coach’s personality motivation component is reflected in the coach’s intrapersonal characteristics, including psychological correlates such as openness, flexibility, empathy, and authenticity. The coach’s love factor essentially reinforces the content of the previous component in those approaches from the athlete’s side. Types of responses that incorporate positive emotions towards the coach and recognition and treatment of the coach are given space here. The coach as a role model is presented as a separate category and refers to the inspiration from the coach and the quality and specificity of the knowledge material the coach passes on, which serves as a model to follow. The respect for authority factor has a narrower content, manifesting itself mainly in the specificity of the role resulting from the hierarchical relationship that characterises formal systems. It is a more inflexible category, where the athlete accepts the specificities of the coach’s role, typically involving behaviour indicative of authoritarianism arising from formalities and formal role systems. In the case of the expectation component, the system of demands from the coach towards the athlete and his/her role is highlighted. It is well related to the desire to conform. It is also in this demand system from the coach’s direction that this can be seen, with coach comparisons made by athletes in some cases. Finally, professionalism appears, which refers to the expertise itself and the transfer of information and knowledge, its content, and the way it is delivered. It also includes consistency and credibility.

## 4. Discussion

The research aimed to explore the microsystemic support factors underlying sport persistence and analyse the role of family, peers, and coaches. To this end, a semi-structured interview survey was conducted, and the responses of 133 athletes were analysed using the grounded theory method.

In terms of *supportive and motivational segments from the family*, six motivational segments were identified in the content analysis about sports activities. Motivational components related to the specific person were dominant factors underlying sport persistence. Family background and sports history played a prominent role in the persistence of the interviewees in the sport. Many play sports to make their parents proud or because sports have always been part of their family life. Parents, especially physical education teachers and professional or amateur sportsmen and -women, encourage their children to play sports by setting an example through perseverance. The support of the family background, parental sports behaviour, and the transmission of sports values strengthen the commitment to sports [13]. In addition, athletes often motivate their family members to adopt a healthy lifestyle and find a common hobby in sports. Those who regularly played sports as children are likelier to remain active later in life [25]. Positive early experiences, competitive success, and recognition from parents all contribute to long-term engagement [26].

For motivation related to family values, health preservation was the main driver. Athletes are primarily concerned with maintaining their health to avoid illness and stay fit. Sport is integral to family values, where regular exercise and perseverance are emphasised. In addition, the values imparted by sport, such as perseverance, discipline, and team spirit, have a positive long-term impact on children’s personality development and social relationships [27,28]. Furthermore, sports are communal activities that allow family members to spend time together, talk to each other, and support each other. Previous research is consistent with the finding that joint sports strengthen family bonds and provide quality time and shared goals, increasing cohesion and support among family members [29,30]. The relaxation effect is also important, as sport helps to relieve tension and contribute to physical and mental well-being. This category reflects well on sports’ physical, mental and social health benefits, which are relevant for both the individual and the family [31]. The love and passion for sport are deeply embedded in the interviewees’ lives, providing them with support and motivation. Mutual support is also significant, where athletes can motivate their family members. This kind of support provides emotional stability. Athletes can share their successes and failures and receive emotional support from their family members, which helps them cope with challenges, strengthens their sense of belonging and cooperation around shared values, and develops their social skills [32,33].

In addition to its present values, sport was also identified as a means to future prosperity. Athletes often consciously prepare for the values they will pass on to their children in the future, which is analogous to the previously presented positive values [15,16]. Sports also offer future livelihood opportunities to support their families financially [21]. Development and parental role modelling have an inspiring effect as parents actively participate in sports and support their children’s goals, which young people who participate in sports can incorporate into their personalities and lifestyles [34]. In addition, subjects sometimes aspired to surpass their parents’ sporting careers and achieve even higher levels of excellence. This may be because their parents’ success may make athletes feel the need to achieve similar or even greater success, or it may also reflect an intrinsic motivation to make their parents happy and proud [26].

The family support system also detected introjected motivational components for persistent sports behaviour. Interviewees set expectations for themselves and strive to meet them through their sporting performance. The desire to prove themselves is a strong driving force, with athletes wanting to elicit pride from their families, which motivates them to persevere in training and competition. Athletes’ goals and expectations help them exercise self-discipline and work consistently to improve. These goals give them direction and help them to continually improve and surpass their previous performance [35].

A specificity of disadvantage compensation motivation emerged, suggesting that sport provides an opportunity to break out of poor circumstances and support talent [36,37]. Separation from family also emerged, suggesting that sport provides an opportunity to break out of the family environment and to be independent. The experiences and challenges gained from playing sports help athletes to become independent. Regular training and competitions require independence, so athletes learn to manage their time and tasks [38].

The *role of peers in sport persistence* can also be of paramount importance. Responses that did not emphasise the importance of peers—either because of individual preferences or the nature of the sport—were categorised separately. Overall, in our research, supportive influences from peers clustered around three main motivational groups, namely individual, community, and relational motivations. Several types of peer support were identified as individual motivation. The desire to prove themselves is well illustrated, where athletes want to show their peers their abilities and performance, which can have a positive impact on both individual achievement and engagement [39]. The desire for individual development is also significant, as peers can help athletes develop personally and achieve maximum performance. The presence of peers can create a competitive environment that encourages athletes to improve their performance [40]. The motivation for health maintenance is directed towards maintaining the physical and mental health of the individual, in which the positive influence of peers can also play an important role. Sports peers are often role models, especially those who follow healthy lifestyle habits such as proper diet, sufficient rest, and stress management. These positive role models can help athletes develop similar habits [41]. Seeking recognition is also an important factor, where athletes value praise and appreciation from peers, which further encourages them to persevere in sports [42].

Community motivations include a collective sense of belonging, emphasising team unity and cultivating friendly relationships. Shared goals and team spirit foster a sense of belonging, which contributes to the emotional well-being of athletes. Commitment to each other and working together has a positive impact on the psychological well-being of athletes [43]. Altruism refers to the support of peers and putting the team’s interests first, in which athletes emphasise that they never leave anyone behind and that team success is more important than individual performance. Altruistic behaviours increase team unity and cohesion, thus also contributing to performance and perseverance [44]. Shared activities and peer motivation are also important, where doing sports together brings joy and helps maintain a commitment to the sport, and a sense of belonging and ‘team as a family’ are also essential factors that help athletes persevere despite difficulties. Belonging to a sporting community provides a sense of security and stability. The community support provided by peers contributes to athletes’ mental and emotional balance, which is essential for continuous improvement [45]. Peer comparison and the desire to compete can also be a motivating force, where athletes compete with each other and with non-athlete peers, as well as peer support, where the support and camaraderie of fellow athletes play a significant role. Belonging to a sporting community provides a sense of security and stability, contributing to athletes’ mental and emotional balance, which is essential for continued development [32]. Finally, a positive environment, with a positive climate and positive feedback and attitudes, is also motivating, as a positive climate makes sports and sporting activities more attractive and increases engagement and persistence [41,46].

The third set of supports covered relational motivations. This includes relationship building, which emphasises the role of teamwork and community building. The resources and relational capital available through sports are also retention factors. Sports peers can help athletes build a wider social network, and the relationships developed in sports often extend beyond the sporting arena and can influence athletes’ careers [47]. Role modelling is also important, where athletes can feel like role models and provide a positive role model for those around them. Peers often serve as positive role models and inspiration for others. In addition, when an athlete sees his or her peers succeeding and working hard, it encourages them to achieve their best and follow effective practices [42]. All of these motivational factors contribute to athletes’ persistence, helping them persevere in sporting activities and maximise their abilities.

Finally, we wanted to explore the *supportive influences of the coach*. A few interviewees said that the coach’s role was irrelevant to them because they play a sport that does not require a coach. There was also a separate category of responses where the coach’s role was not emphasised for individual or sport-specific reasons. Apart from these specific cases, however, a significant supportive effect was detected in the research, which in this case could also be classified into several types. General support ranged from average to full support, in which interviewees were unable or unwilling to specify the nature of the support.

In individual motivation, the role of the coach is to develop and recognise the athlete’s talent. For athletes, recognition by the coach is essential to support their self-esteem, confidence, and performance, thereby increasing their intrinsic motivation and perseverance [48]. The motivation for health maintenance lies in following the patterns of health-conscious behaviour demonstrated by the coach, which can become an intrinsic need for athletes that can also increase performance and endurance [27]. Development as a motivational component includes the practical application of knowledge imparted by the coach, the coach’s love for the athlete, and ethical and psychological aspects [49].

Community motivation includes factors that emphasise the central role of peers and the community. This segment reflects well the prioritisation of team interests over individual interests. If the athlete feels that he or she is recognised and supported by his or her coach, he or she is more likely to accept the coach’s advice and instructions, either for individual or team-related behaviour [50]. Altruism can also be included here, which refers to team spirit and mutual support, in which the coach as team leader is given a prominent role [51] in maintaining and reinforcing a sense of belonging to something.

Motivation by the person of the coach was the largest category, comprising several subgroups. Coach personality is a motivational component that reflects intrapersonal characteristics such as openness, flexibility, empathy, and authenticity. An empathic and open-minded coach creates an environment where the athlete feels safe and valued. This increases the athlete’s trust in the coach, which is key to engagement. In addition, the inspirational personality of the coach encourages the athlete to strive for his/her best performance and to commit to the sport in the long term [52]. The coach’s love and appreciation (from the athlete) also play an essential role in motivating athletes. If the athlete feels that the coach cares about them, not only as an athlete but also as a person, this increases engagement from both directions [13]. An authoritarian leadership style and respect for authority, stemming from formal hierarchical relationships, have also been reported, and to an optimal extent, this can also contribute to engagement and persistence [14]. This is well matched by the coach expectancy component, which refers to a set of demands from the coach’s direction that can increase athletes’ desire to comply [53]. Finally, professionalism, which refers to the coach’s expertise and information transfer skills, also plays an important role in coach credibility and motivation of athletes [54].

## 5. Conclusions

In the context of sports, the role of family, peers, and coaches as sources of support is significant. The role of family support and motivation in the continuation of sporting activities is significant. The actions of parents, their own sporting history, and the sporting values they instil in their children serve to encourage athletes. It is a common occurrence for athletes to desire to make their parents proud, with sports forming an integral part of their family life. Concerning the aspect of the family, our study emphasised the positive impacts of disadvantage compensation, introjected (family-related) motivation, person-related support, the support related to family values, and support for the future.

Furthermore, the influence of fellow athletes is a crucial factor in determining an athlete’s ability to persevere and perform at their optimal level. In our study, we could contextualise the most relevant influential factors around community, individual, and relational motivation. Regarding individual motivation, peer support has been demonstrated to enhance athletes’ self-confidence and performance. Fostering altruism and team spirit can contribute to developing a sense of belonging, while competitive situations can also serve as a source of motivation. The example of a healthy lifestyle set by fellow athletes can assist in maintaining the health of athletes. Relational motivations, such as the formation of communities and the provision of peer support, contribute to athletes’ emotional and mental well-being. Moreover, the relationships formed through participation in sports provide support in other areas of life in addition to the context of sports.

The supportive role of the coach can also be demonstrated, despite some feeling that it is less relevant in certain types of sports. The provision of general support, talent development, and coach recognition has been demonstrated to enhance athletes’ self-esteem and performance. We could detect the influencing roles of community motivation, individual motivation, and motivation by the coach’s behaviour. The coach’s demonstration of health-conscious behaviour and the transfer of knowledge and norms facilitate the development of athletes. In terms of social motivation, the cultivation of team spirit and altruism are important. The coach’s personality, empathy, openness, and professional credibility enhance the athletes’ confidence, commitment, and motivation in sports.

As a limitation, we should note that the sample is not representative, which means that the results cannot be generalised to the entire population of athletes. It is also noteworthy that the interviewees frequently encountered difficulty differentiating between motivation and perseverance. This highlights the need for further research to establish a clear distinction between these two concepts. We should emphasise that the data derived from interviews are based on self-reported answers that introduce potential issues like social desirability bias or recall bias, especially in questions related to motivations and family influences. The lack of triangulation methods (e.g., comparing interview data with observational data or other sources) may also be mentioned as a limitation.

Nevertheless, the research has considerable practical implications that may impact both the athlete and their surrounding environment. It would be beneficial for coaches and mentors to understand the individual motivations and factors that support commitment and perseverance to provide more effective guidance and assistance to their athletes in their development. Furthermore, the results can inform the development of training techniques and motivational approaches. Studies have demonstrated that the momentum of athletes is influenced by a range of factors, including personal growth and community involvement. Effective training programmes must, therefore, consider these variables. Additionally, the findings emphasise the significance of community involvement and social recognition as crucial motivational elements. Consequently, sports clubs and organisations can benefit from implementing initiatives and activities that foster a sense of community and recognition. Future research could apply quantitative methods to test the findings and theoretical categories developed in this qualitative study, and a longitudinal approach could be used to track sport persistence over time. Since this study focuses on national culture and traditions (macrosystem), future research could involve cross-cultural comparisons to examine how sport persistence differs in various countries or regions.

## Figures and Tables

**Figure 2 jfmk-09-00196-f002:**
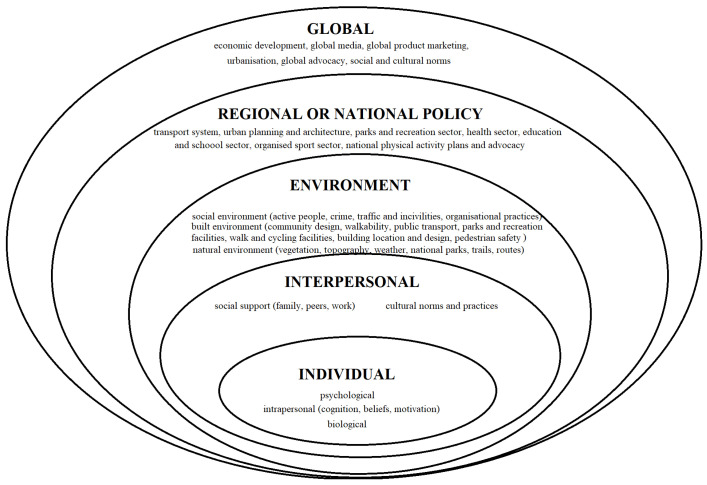
Bauman’s ecological model adapted for sports [8].

**Figure 3 jfmk-09-00196-f003:**
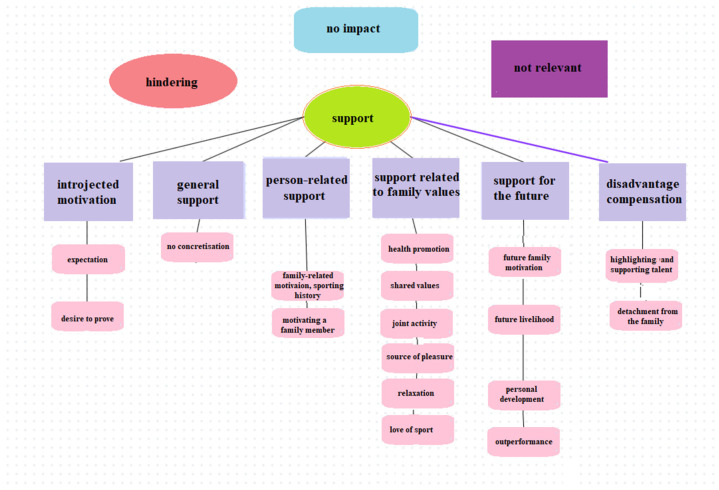
Family-related factors influencing sport persistence.

**Figure 4 jfmk-09-00196-f004:**
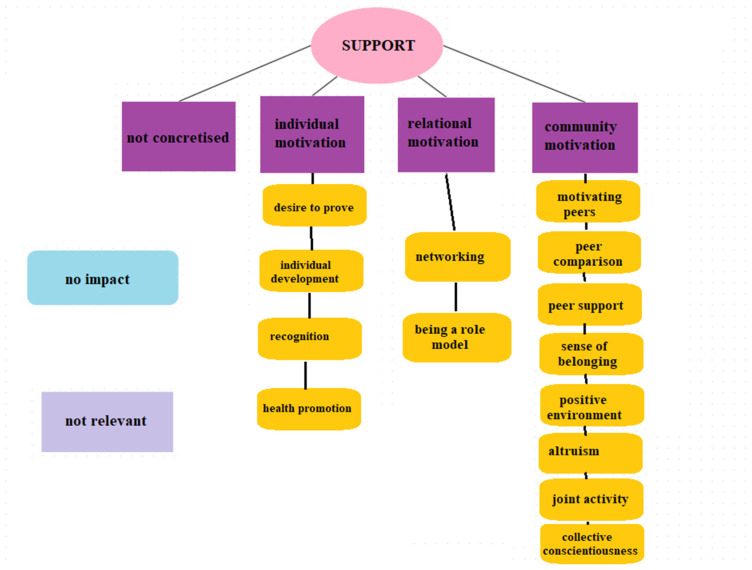
Peer-related factors influencing sport persistence.

**Figure 5 jfmk-09-00196-f005:**
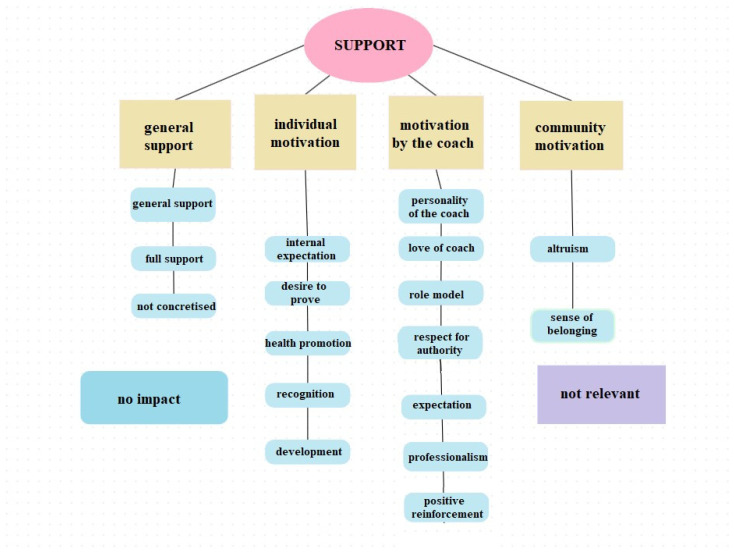
Coach-related factors influencing sport persistence.

**Table 1 jfmk-09-00196-t001:** Sample composition by gender, parental education, level of education, type of sport, and level of sport participation.

	N	%
**Gender**		
male	80	60.2
female	53	39.8
**Mother’s education**		
primary	10	7.5
secondary	71	53.4
higher level	52	39.1
**Father’s education**		
primary	11	8.3
secondary	81	60.9
higher level	41	30.8
**Own level of education**		
secondary	85	63.9
tertiary	48	36.1
**Type of sport**		
team sports	82	61.7
individual sports	51	38.3
**Sporting level**		
as a leisure activity	41	30.8
I also participate at a competitive level	92	69.2

**Table 2 jfmk-09-00196-t002:** The script of the interview.

Dimension	Variables	Question
**Sociodemographic background**	**age, gender, parents’ education, parents’ labour market status, subjective financial situation, educational level**	How old are you? What is your sex? What is the highest educational attainment of your mother/father? How could you evaluate your family’s financial status? Is your mother/father employed? What is the level of your current studies?
**Sport-specific questions**	**sport frequency, sport type, sport level, career stages**	How often do you pursue sport? What type of sport do you play most often? Which type of sport do you pursue? At what level do you pursue sport? When did you started to pursue a sport? What kind of stages could you determine in your sports biography?
**Competitive activities**	**activities other than sport (e.g., study, work, other competitions**	Besides sport, what kind of other activities could you mention in your life? How can you evaluate your school life and academic achievement? Have you ever been employed/are you employed? Do you pursue any other activities at a high level (e.g., playing a musical instrument)?
**Individual motivation**	**personal drives**	Why do you pursue sport? Why are you persistent in pursuing sport? Why do you persist with your sporting activities?
**Sport persistence—microsystem**	**family** **peers** **coach**	If you consider only your family (excluding other factors), why do you continue to do sports? If you consider only your peers (excluding other factors), why do you continue to do sports? If you consider only your coach (excluding other factors), why do you continue to do sports?
**Sport persistence—mesosystem**	**family–school relationship** **family–sports–peers relationship**	How does the relationship between your family and your school contribute to your perseverance in sport? How does the relationship between your family and your sporting teammates contribute to your perseverance in sport?
**Sport persistence—exosystem**	**sporting environment**	If you only consider your sporting environment, e.g., infrastructure (excluding other factors), why do you continue to do sports
**Sport persistence—macrosystem**	**national culture and traditions**	If you only consider national culture and traditions (excluding other factors), why do you continue to do sports?

**Table 3 jfmk-09-00196-t003:** Factors underlying sport persistence in terms of the role of the family.

Main Categories	Sub-Categories	Examples
**disadvantage compensation**	**highlighting and supporting talent** **(n = 1)**	“*It was my foster parents who saved me, they were the ones who saw the talent in me.*” (Participant No. 54)
	**detachment from the family** **(n = 1)**	“*Since I live at home, we are together a lot. I can get away from home in the meantime.*” (Participant No. 119)
**introjected motivation**	**expectation** **(n = 2)**	“to make *them proud of me*” (Participant No. 44)
		“*to make my parents proud of me*” (Participant No. 45)
	**desire to prove** **(n = 14)**	“*because no one in my family plays sport at the level I do and it’s nice, when they watch me, cheered on me and are proud of me.*” (Participant No. 40)
		“*I would like to make them proud of me*” (Participant No. 118)
		“*they are happy when they see a performance and it’s good to see them happy and proud.*” (Participant No. 123)
**person-related support**	**family-related motivation, sporting history** **(n = 15)**	“*because several members of my family played the sport*” (Participant No. 39)
		“*I come from a sporty family, sport has been part of my everyday life since childhood*” (Participant No. 52)
		“*Almost everyone on my dad’s side is involved in sport, so it was really through his influence that I started and have continued ever since.*”
		(Participant No. 142)
	**motivating a family member** **(n = 15)**	“*my mother teaches physical education, she motivates me*” (Participant No. 12)
		“*to encourage them to stay healthy and to have hobbies we can do together.*” (Participant No. 81)
		“*because I can motivate them that no matter how bad my physical condition is (due to injuries), I won’t give up.*” (Participant No. 115]
**support for the future**	**future family** **motivation** **(n = 1)**	“*because I also want my children to play sport in the future.*” (Participant No. 120)
		“*because I love it and I want my family to play sport when I have children*” (Participant No. 133)
	**future livelihood (n = 3)**	“*I want to make a living from it later*” (Participant No. 23)
		“*I will even be able to spend the money on my family*” (Participant No. 27)
		“*so that I can provide them with opportunities later on.*” (Participant No. 46)
	**personal development** **(n = 6)**	“*they are very supportive, they do their best to help me achieve my goals and improve*” (Participant No. 48)
		“*get as much out of life as possible as a professional athlete*” (Participant No. 109)
		“*I think my parents are a perfect example of if you don’t get out and you’re not part of a good community or you’re not playing sports, you don’t have anything that makes you feel good and you’re just in this dark kind of monotonous life, so you’re working, you’re in a job you don’t like, you’re in a school you don’t like, it can make you go crazy, so you get really sad then. So they motivate me to do everything I can while I’m young, to have friends, to go to the gym and everything, to not be like them so to speak.*” (Participant No. 134)
	**outperformance** **(n = 1)**	“*because I want to reach a higher level in sport than them*” (Participant No. 43)
**support related to family values**	**health promotion (n = 2)**	“*to avoid diseases from which my relatives have suffered and are suffering*” (Participant No. 63)
		“*because my family is a bit overweight and I know how difficult it is to live like that*” (Participant No. 86)
		“*healthy living, living as long as possible with loved ones in good health*” (Participant No. 124)
	**shared values** **(n = 8)**	“*because sport is important to my family*” (Participant No. 106)
		“*they also think regular exercise is important*” (Participant No. 111)
		“*I was brought up from an early age never to give up*” (Participant No. 130)
	**joint activity (n = 3)**	“*they contribute well. And so we can talk, spend time together.*” (Participant No. 4)
		“*it could be a joint programme*” (Participant No. 9)
		“*they know I like doing it, they always ask me about my progress or where I am at*” (Participant No. 49)
	**source of pleasure** **(n = 14)**	“*I want to see them proud*” (Participant No. 28)
		“*my family supports me in this, and it is a source of joy for them to see me happy and getting on in life and doing what I want to do. And that motivates me.*” (Participant No. 137)
	**relaxation** **(n = 1)**	“*tension relief*” (Participant No. 26)
		“*because it helps me unwind*” (Participant No. 84)
		“*because this is what helps me relax and gives me happiness*” (Participant No. 136)
	**love of sport** **(n = 3)**	“*I love it and it is part of my life*” (Participant No. 34)
		“*I have a relative and we know very well that he loves riding and he gives me a motivation because you can see that he really enjoys it and takes care of horses as well.*” (Participant No. 135)

**Table 4 jfmk-09-00196-t004:** Factors underlying sport persistence in terms of the role of peers.

Main Categories	Sub-Categories	Examples
**community motivation**	**collective conscientiousness (n = 5)**	“*For the sake of team unity, cultivating and maintaining friendships.*” (Participant No. 38) “*It’s because we are a team and if one person leaves, the team is no longer complete.*” (Participant No. 43) “*There were many times when I did it just for them, so as not to let them down*” (Participant No. 132)
	**altruism (n = 7)**	“*Because I’m not going to leave them in the middle of a tournament.*” (Participant No. 29) “*Because I know that they need me and they count on me in the team and because I like to play sports with them.*” (Participant No. 40) “*We’ve built a good team over the years and it would be a shame to leave that environment. I don’t want to let them down.*” (Participant No. 123)
	**joint activity (n = 8)**	“*A common point for conversation, a good and meaningful time with friends.*” (Participant No. 27)
		“*Because together these times are good times.*” (Participant No. 35)
		“*They motivate me and they play the same sport and I enjoy my time with them.*” (Participant No. 48)
	**motivating peers** **(n = 2)**	“*I want to motivate them myself.*” (Participant No. 69) “*I motivate them to achieve better results*. ” (Participant No. 70) “*Because I can encourage them to take up sport themselves.*” (Participant No. 87)
	**peer comparison** **(n = 7)**	“*Because I want to be better than them, and I like being envied.*” (Participant No. 75) “*To be accepted by them.*” (Participant No. 78) “*It’s uplifting to be able to do better among people who spend most of their free time just staying at home, hanging out, partying. With less free time, it’s hard to reach a level you’re happy with. And if I take my fellow teammates into acccount, I am happy if I can show them one or two things by example or if I can get their attention and recognition by my will, attention and performance.*” (Participant No. 116)
	**peer support (n = 40)**	“*It’s about sticking together and fighting for each other.*” (Participant No. 52) “*I like my teammates, there’s always a good atmosphere.*” (Participant No. 60) “*Because I play sport with my friends and they support me to keep going.*” (Participant No. 86) “*My circle of friends and my friends are here on the farm, the people I ride with and compete with, and they always support each other, we talk about everything, and they’re important to us, so they always add to the atmosphere, so that’s why.*” (Participant No. 137)
	**sense of belonging (n = 5)**	“*I see my team as a second family.*” (Participant No. 33) “*I really like my team, and most of my friends are athletes anyway.*” (Participant No. 83) “*It’s a good community, I like it here, and I have many friends I’ve met through horse riding here.*” (Participant No. 139)
	**positive environment (n = 7)**	“*I am in a supportive environment with people who are persistent.*” (Participant No. 16) “*The environment and atmosphere is motivating.*” (Participant No. 102) “* They motivate me, encourage me, praise me all the time, everyone is positive towards me and the circumstances are positive.*” (Participant No. 131)
**individual motivation**	**desire to prove (n = 3)**	“*To show that I am somebody.*” (Participant No. 31) “*I want to prove myself to them.*” (Participant No. 103)
	**individual** **development** **(n = 11)**	“*It’s uplifting to be able to do better among people who spend most of their free time just hanging out, hanging out, partying.*” (Participant No. 19) “*With less free time, it’s difficult to reach a level you’re happy with. And if I consider my fellow coaches, I am happy if I can show them one or two things by example or if I can get their attention and recognition by my will, attention and performance.*” (Participant No. 116) “*I’m doing my best, but we play as a team.*” (Participant No. 54) “*To get as much out of life as possible as an athlete.*” (Participant No. 109)
	**health** **promotion** **(n = 2)**	“*For my health*” (Participant No. 76) “*To get fit, that’s what they motivates me to achieve*” (Participant No. 80)
	**recognition (n = 5)**	“*I like it when others praise my performance.*” (Participant No. 11)
		“*I like to be with my teammates and I like to be praised for being a good goalkeeper.*” (Participant No. 25)
		“*To be seen for who I am and to be respected and recognised. I want them to be proud of me.*” (Participant No. 68)
**relational motivation**	**networking** **(n = 5)**	“*It builds team spirit and community.*” (Participant No. 84) “*Because I have a lot of friends from there, so that’s when I meet them. These relationships are important in my life.*” (Participant No. 119) “*Through sport you can meet a lot of people, make a lot of friends. And some people are motivated by it, some people are not, but I am.*” (Participant No. 143)
	**being a role model** **(n = 2)**	“*…because they might take me as an example.*” (Participant No. 5) “*I want to set an example to them and they motivate me.*” (Participant No. 55) “*Because I can set an example for them by having a willingness to train that others would like to have.*” (Participant No. 115)

**Table 5 jfmk-09-00196-t005:** Factors underlying sport persistence in terms of the role of the coach.

Main Categories	Sub-Categories	Examples
**community motivation**	**altruism (n = 3)**	“*Because I would not disappoint them and help them.*” (Participant No. 99) “*It helps to keep the group together*” (Participant No. 126) “*Well, because I’ve developed such a relationship with the coach that whatever happens, I wouldn’t let the team down, I wouldn’t let the coach down.*” (Participant No. 140)
	**sense of belonging (n = 5)**	“*To contribute to the team.*” (Participant No. 27) “*He is looking after the interests of the team.*” (Participant No. 127)
**individual motivation**	**internal expectation** **(n = 3)**	“*Because I want to see that talent in myself*” (Participant No. 45) “*To make you them proud of me*” (Participant No. 78)
	**desire to prove (n = 7)**	“*I want to prove that I can indeed do anything.*” (Participant No. 28) “*To show that I am persistent*” (Participant No. 82) “*I want to show my old coaches how much I’ve improved, and I don’t want to let my current ones down.*” (Participant No. 113)
	**health promotion (n = 4)**	“*…because it does a lot to keep me healthy.*” (Participant No. 5) “*…supports me in living a healthy lifestyle.*” (Participant No. 69)
	**recognition (n = 2)**	“*I want them to be proud of me.*” (Participant No. 118) “*To get his attention.*” (Participant No. 119)
	**development** **(n = 19)**	“*He motivates me, prepares me well, and has the knowledge and experience for the sport.*” (Participant No. 48) “*It not only teaches me to be a better player but also helps me to become a better person.*” (Participant No. 52)
**motivation by the coach**	**personality of the coach (n = 7)**	“*We have a good coach, sometimes strict, as it should be.*” (Participant No. 60) “*My coach saw my potential; he thinks I can have a future in this sport and supports me.*” (Participant No. 114) “*…I’ve been riding in several places, so I’ve observed several trainers and how they teach, treat the horses, and relate to people. And my trainer was a very, very pleasant positive disappointment because I thought all trainers were a little bit like that, a little bit more temperamental, or well, they don’t really care about the kids’ feelings, and they really just focus on the sport, but she’s totally not like that.*” (Participant No. 134)
	**love of coach (n = 5)**	“*Because he is the best coach.*” (Participant No. 18) “*I really like our coach*” (Participant No. 91) “*Best coach in the world!*” (Participant No. 104)
	**role model (n = 3)**	“*He does it at a very high professional level; he’s a role model.*” (Participant No. 105) “*Well, actually, there’s probably one coach at the moment that I look up to and who influences me, well he’s the one that I look up to, and precisely because of the way that I look up to him, the training sessions that he has and it’s definitely worth going to. And it’s not that you want to be like him, but it’s just that you’re probably not stupid if that’s where he’s positioned himself.*” (Participant No. 141)
	**respect for authority (n = 4)**	“*I respect and appreciate his authority as a competent person in his profession, and that’s really all.*” (Participant No. 142) “*I respect him too; I look up to my coach.*” (Participant No. 143)
	**positive reinforcement (n = 7)**	“*I like it when he praises me.*” (Participant No. 25) “*He always only encourages me*” (Participant No. 83) “*Because it gives a lot of strength and encouragement.*” (Participant No. 88) “*He counts on me, he is proud of me, he motivates me.*” (Participant No. 131)
	**expectation (n = 6)**	“*No matter how strict the coach is, you must stick it out because he’s not coaching for nothing.*” (Participant No. 4) “*I have worked with several coaches in my sporting career, and I would like to thank them for their work by saying that I do my job to the best of my ability*” (Participant No. 128) “*Because the coach also has a lot of work to do. And then I know that I have to do the same, and then all the work that I put in, that we put in together, doesn’t go to waste.*” (Participant No. 132)
	**professionalism (n = 6)**	“*They have very good professional skills.*” (Participant No. 34) “*He’s very consistent and has good training sessions.*” (Participant No. 39) “*Because the coach can design the right training plan for me.*” (Participant No. 87) “*For the transfer of professional knowledge.*” (Participant No. 108)

## Data Availability

Data are available only on request due to ethical restrictions.

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
