# Peer review of "Qualitative Analysis of Micro-System-Level Factors Determining Sport Persistence"

_jfmk, 2024, doi:10.3390/jfmk9040196_

Round 1

Reviewer 1 Report

Comments and Suggestions for Authors

First of all, we would like to thank the authors for the valuable research they have carried out.

In the attached PDF document, some comments that should be considered have been added. 

Other important considerations are the following:

The introduction lacks depth and needs to be improved by considering other studies that more solidly support the purpose of the research.

The method does not describe the procedures, the analysis of results with ATLAS.ti software, where it would be necessary to add some diagrams or figures that explain more clearly the object of study.

Considering that the type of qualitative study and the use of grounded theory must move in flexible environments, in terms of facing unpredictable fields of action. For this reason, the research was structured in the following phases (example: phase 1): definition of the sample; phase 2): Validation of the interview script; 3): Application of the interviews; 4): Transcription and analysis; 5): coding and categorization; 6): Interpretation.

Describe each of these steps or those required by the research group to support its study in greater depth.

It would be necessary to detail the quality criteria followed for the present research. For example: “the present study resorted to the following quality criteria, exposed by (Flick, 2012; Flores, 2009): Credibility; data-rich elicitation; commitment to fieldwork; confirmability.”

It is suggested to review the previous example in order to substantiate the methodological rigor of the study.

The results should be better organized to articulate the shared text with the textual fragments of the tables.

The discussion is well elaborated.

The conclusions should be revised and seek to respond to the objective of the study.

It is important to add a section on limitations, as well as future perspectives, recommendations or practical applications.

Finally, review the suggestions made in the tables and references.

I thank the authors again and encourage them to review each of the comments made.

Author Response

Thank you very much for your supportive review. Based on your comments, the following modifications were carried out:

Reviewer1: First of all, we would like to thank the authors for the valuable research they have carried out. In the attached PDF document, some comments that should be considered have been added. 

Authors: Thank you very much for the positive and supportive comments, we incorporated the suggestions presented in the PDF file.

Reviewer1: The introduction lacks depth and needs to be improved by considering other studies that more solidly support the purpose of the research.

Authors: Authors: Thank you very much for your kind notification. The literature review was extended with further relevant resources.

Reviewer1: The method does not describe the procedures, the analysis of results with ATLAS.ti software, where it would be necessary to add some diagrams or figures that explain more clearly the object of study.

Authors: Thank you very much for your kind notification and suggestion. We would like to note that the software was not used for creating figures since we aimed to provide another kind of visual information. Therefore, we created our figures in Canva program. Also, to support the better understanding of the interview’s structure, the dimensions of the interview are presented in Table 2.

Reviewer1: Considering that the type of qualitative study and the use of grounded theory must move in flexible environments, in terms of facing unpredictable fields of action. For this reason, the research was structured in the following phases (example: phase 1): definition of the sample; phase 2): Validation of the interview script; 3): Application of the interviews; 4): Transcription and analysis; 5): coding and categorization; 6): Interpretation. Describe each of these steps or those required by the research group to support its study in greater depth.

Authors: Thank you very much for your kind comment. We gave a pint-by-point description for the introduction of this information.

Reviewer1: It would be necessary to detail the quality criteria followed for the present research. For example: “the present study resorted to the following quality criteria, exposed by (Flick, 2012; Flores, 2009): Credibility; data-rich elicitation; commitment to fieldwork; confirmability.” It is suggested to review the previous example in order to substantiate the methodological rigor of the study.

Authors: Thank you very much for your kind suggestion. We reviewed the references you mentioned and applied the content related to the methodology of the current paper.

Reviewer1: The results should be better organized to articulate the shared text with the textual fragments of the tables.

Authors: Thank you very much for your kind notification. We tried to better organise the content of the Results section.

Reviewer1: The discussion is well elaborated.

Authors: Thank you very much for your kind feedback.

Reviewer1: The conclusions should be revised and seek to respond to the objective of the study.

Authors: Thank you very much for your kind suggestion. We modified the Conclusions following your advice.

Reviewer1: It is important to add a section on limitations, as well as future perspectives, recommendations or practical applications.

Authors: Thank you very much for your kind suggestion. The limitations and practical implications at the end of the Conclusions section were revised and completed. Future perspectives were also added.

Reviewer1: Finally, review the suggestions made in the tables and references.

 Authors: Thank you very much for your conscientious work!

We hope that we could improve the quality of the manuscript and it can be accepted for publication.

Sincerely,

the Author

Reviewer 2 Report

Comments and Suggestions for Authors

  • A brief summary: This research explored sport persistence, which encompasses the ability to maintain performance despite challenges such as plateaus, failures, and injuries. Utilizing qualitative methods grounded in Bronfenbrenner's socio-ecological model, the study involved semi-structured interviews with 133 high school and university athletes. Data analysis, facilitated by Atlas.ti and Grounded Theory, revealed that family, peers, and coaches significantly influence sport persistence at the micro-system level. Various motivational components were identified for each group, including family values and individual factors related to peers and coaches. The findings highlight the need for targeted interventions that address these systemic factors to promote youth sports and physical activity, ultimately fostering hope and motivation for the future of young athletes.
  • General concept comments
    Article: This article exhibits several weaknesses, particularly in its English translation and grammar, which occasionally hinder clarity and comprehension. Additionally, the text would benefit from the inclusion of more figures to visually represent key concepts and data, enhancing overall understanding and engagement for readers. Addressing these issues could significantly improve the article's effectiveness and accessibility.

Review: The topic of this article is intriguing and has the potential to greatly benefit the readership. However, there are some gaps in the content that may leave certain aspects underexplored for specific audience segments. To fully address these gaps and enhance the article's relevance, I recommend examining the specific comments for more detailed insights and suggestions. This would help ensure the material resonates more effectively with all readers.

  • Specific comments:
    • Lines 45-47: the word “commitment” is used three times in one sentence; reconsider this sentence; throughout your entire manuscript, when listing multiple attributes, for example, consider listing them in alphabetical order, as it will be easier for the reader to follow (e.g., adaptive coping mechanisms, positive personality traits, and resilience)
    • Line 50: clarify what “this” is use a comma after “process”
    • Lines 51-56: the English grammar and syntax need to be reorganized, as the sentences are difficult to follow. I would encourage caution throughout the manuscript of repeating the same word in a sentence (e.g., Line 55 and “performance”)
    • Line 57: grammatically incorrect
    • Line 64: comma after “attitudes”
    • Line 70: comma after “exo”
    • Line 71:  consider starting the sentence with your allusion to a Matryoshka doll set
    • Line 73: for this sentence, and those mentioned before and throughout the rest of the text, you will need to review to understand if you need to add a comma before the word “and” when you are listing words.
    • Lines 65-82: you may want to consider a figure or table that describes the Bronfenbrenner’s model
    • Line 100: is there a reason to describe Bauman et al.’s model?
    • Line 121: Because there are societal influences, it could be worth noting in what country and/or region of a country the participants are from.
    • Table 1: consider using “male” and “female”, instead of “boy” and “girl”; I am also unsure who you are describing in the “level of education numbers, is it the participants? Would it be worthwhile to have all of the participants information listed first, then followed by the parents?
    • Table 1:  you may consider stating that it is the parents’ highest level of education attained.
    • Line 144: there are double parentheses
    • Line 159: where were the interviews done? How were they transcribed?
    • Line 187:  I would recommend mentioning the Table 2 with the results in this paragraph
    • Line 216: should this be a new paragraph?
    • Table 2: what are the bracket numbers at the end of each quote (participant id)? There are also words that have a line striking them (i.e., love of sport and mutual support sections); also make sure that italics are used consistently (i.e., introjected motivation quote 44).
    • Line 257: there should be mention of Table 3 in this paragraph
    • Line 299: there should be mention of Table 4 in this paragraph
    • Line 355: consider rephrasing the “coach’s person” as it does not read well
    • Line 420: why is it in italics? There are other similar times when italics seem out of place
    • References: make sure that there is consistence with how journals are listed (i.e., full name of journal vs. abbreviated name)

General questions to help guide your review report for research articles:

  • Is the manuscript clear, relevant for the field and presented in a well-structured manner? There are some issues with English grammar. Additionally, there is no context as to where this research took place.
  • Are the cited references mostly recent publications (within the last 5 years) and relevant? The cited references are well distributed. Does it include an excessive number of self-citations?
  • Is the manuscript scientifically sound and is the experimental design appropriate to test the hypothesis? The qualitative methodology appears to be well documented and the use of software for thematic analysis is relevant.
  • Are the manuscript’s results reproducible based on the details given in the methods section? Because it is a qualitative study, I do not know if the results are reproducible, but the method can be reproduced.
  • Are the figures/tables/images/schemes appropriate? Do they properly show the data? Are they easy to interpret and understand? Is the data interpreted appropriately and consistently throughout the manuscript? Please include details regarding the statistical analysis or data acquired from specific databases. More figures are recommended.
  • Are the conclusions consistent with the evidence and arguments presented? Yes, the conclusions appear to match the evidence.
  • Please evaluate the ethics statements and data availability statements to ensure they are adequate. The authors state the study was funded but does not appear to have any ethical implications. The authors may need to edit the “Informed Consent Statement” as it looks like it was just a cut and paste.

7.5. Rating the Manuscript

Comments on the Quality of English Language

Some grammatical issues disrupt the flow for the reader.

Author Response

Thank you very much for your supportive review. Based on your comments, the following modifications were carried out:

Reviewer 2: A brief summary: This research explored sport persistence, which encompasses the ability to maintain performance despite challenges such as plateaus, failures, and injuries. Utilizing qualitative methods grounded in Bronfenbrenner's socio-ecological model, the study involved semi-structured interviews with 133 high school and university athletes. Data analysis, facilitated by Atlas.ti and Grounded Theory, revealed that family, peers, and coaches significantly influence sport persistence at the micro-system level. Various motivational components were identified for each group, including family values and individual factors related to peers and coaches. The findings highlight the need for targeted interventions that address these systemic factors to promote youth sports and physical activity, ultimately fostering hope and motivation for the future of young athletes.

Authors: Thank you very much for your kind feedback.

Reviewer 2: General concept comments. Article: This article exhibits several weaknesses, particularly in its English translation and grammar, which occasionally hinder clarity and comprehension. Additionally, the text would benefit from the inclusion of more figures to visually represent key concepts and data, enhancing overall understanding and engagement for readers. Addressing these issues could significantly improve the article's effectiveness and accessibility.

Authors: Thank you very much for your kind feedback. We tried to address each issue.

Reviewer 2: The topic of this article is intriguing and has the potential to greatly benefit the readership. However, there are some gaps in the content that may leave certain aspects underexplored for specific audience segments. To fully address these gaps and enhance the article's relevance, I recommend examining the specific comments for more detailed insights and suggestions. This would help ensure the material resonates more effectively with all readers.

Authors: We would like to express a special thanks to your detailed comments. We believe it was a significant support in improving the manuscript.

Reviewer 2: Lines 45-47: the word “commitment” is used three times in one sentence; reconsider this sentence; throughout your entire manuscript, when listing multiple attributes, for example, consider listing them in alphabetical order, as it will be easier for the reader to follow (e.g., adaptive coping mechanisms, positive personality traits, and resilience)

Authors: Thank you very much for the suggestion. We modified the sentence and reconsidered the list of attributes when it was possible.

Line 50: clarify what “this” is use a comma after “process”

Authors: Thank you very much for the suggestion, we modified the sentence.

Reviewer 2: Lines 51-56: the English grammar and syntax need to be reorganized, as the sentences are difficult to follow. I would encourage caution throughout the manuscript of repeating the same word in a sentence (e.g., Line 55 and “performance”)

Authors: Thank you very much for the suggestion, we modified the sentences.

eviewer 2: Line 57: grammatically incorrect

Authors: Thank you very much for the suggestion, we tried to correct the sentence.

Reviewer 2: Line 64: comma after “attitudes”

Authors: Thank you very much for the suggestion, we modified the sentences.

Reviewer 2: Line 70: comma after “exo”

Authors: Thank you very much for the suggestion, we placed the comma.

Reviewer 2: Line 71:  consider starting the sentence with your allusion to a Matryoshka doll set

Authors: Thank you very much for the suggestion, we reorganised the sentences.

Reviewer 2: Line 73: for this sentence, and those mentioned before and throughout the rest of the text, you will need to review to understand if you need to add a comma before the word “and” when you are listing words.

Authors: Thank you very much for the suggestion, we reorganised the sentences.

Reviewer 2: Lines 65-82: you may want to consider a figure or table that describes the Bronfenbrenner’s model

Authors: Thank you very much for the suggestion, we added a figure summarising the model.

Reviewer 2: Line 100: is there a reason to describe Bauman et al.’s model?

Authors: Thank you very much for your question. In one of our previous articles focusing on sport persistence, one of our reviewers suggested us to use Bauman et al’s model instead of Bronfenbrenner’s model since Bauman et al modified the original ecological model of Bronfenbrenner to the context of sport. Therefore, it is worth using this model as a theoretical conceptualisation. However, we believe that the original model should also be emphasised due to its fundamental but general content. This is the reason why we use both models.

Reviewer 2: Line 121: Because there are societal influences, it could be worth noting in what country and/or region of a country the participants are from.

Authors: Thank you very much for the suggestion, we specified the country of origin of the participants.

Reviewer 2: Table 1: consider using “male” and “female”, instead of “boy” and “girl”; I am also unsure who you are describing in the “level of education numbers, is it the participants? Would it be worthwhile to have all of the participants information listed first, then followed by the parents? You may consider stating that it is the parents’ highest level of education attained.

Authors: Thank you very much for the notification. We changed the terms ‘boy’ and ‘girl’, and also clarified the parents’ and own level of education.

Reviewer 2: Line 144: there are double parentheses

Authors: Thank you very much for the notification. We corrected it.

Reviewer 2: Line 159: where were the interviews done? How were they transcribed?

Authors: Thank you very much for the suggestion, we deleted the symbol.

Reviewer 2: Line 187:  I would recommend mentioning the Table 2 with the results in this paragraph

Authors: Thank you very much for the suggestion, we referred to Table 2 in the text.

Reviewer 2: Line 216: should this be a new paragraph?

Authors: Thank you very much for the suggestion, we carried out the modification.

Reviewer 2: Table 2: what are the bracket numbers at the end of each quote (participant id)? There are also words that have a line striking them (i.e., love of sport and mutual support sections); also make sure that italics are used consistently (i.e., introjected motivation quote 44).

Authors: Thank you very much for your kind notifications. These numbers referred to the ID number of the participants who gave the given quote. We modified these numbers to ‘Participant No.’ in the whole text.

Reviewer 2: Line 257: there should be mention of Table 3 in this paragraph

Authors: Thank you very much for the suggestion, we referred to Table 3 in the text.

Reviewer 2: Line 299: there should be mention of Table 4 in this paragraph

Authors: Thank you very much for the suggestion, we referred to Table 3 in the text.

Reviewer 2: Line 355: consider rephrasing the “coach’s person” as it does not read well

Authors: Thank you very much for the notification, we modified the heading.

Reviewer 2: Line 420: why is it in italics? There are other similar times when italics seem out of place

Authors: Thank you very much for the suggestion. We wanted to highlight the three main factors investigated in the study i.e. family, peers and coach and we chose to write these terms in italics when we prepared the manuscript.

Reviewer 2: References: make sure that there is consistence with how journals are listed (i.e., full name of journal vs. abbreviated name)

Authors: Thank you very much for the notification. We used Zotero software to ensure the consistent and correct use of references.

We hope that we could improve the quality of the manuscript and it can be accepted for publication.

Sincerely,

the Author

Reviewer 3 Report

Comments and Suggestions for Authors

Thank you for the opportunity to review this manuscript.

The manuscript addresses an under-researched area of sport persistence using a qualitative approach grounded in Bronfenbrenner's socio-ecological model. This is a novel approach, as much of the existing literature focuses on quantitative measures of sporting habits and motivation. By focusing on micro-system level factors, the study provides new insights into the specific roles of family, peers, and coaches in the persistence of young athletes in sports. However, the manuscript lacks a clearer articulation of how its findings differ from or challenge existing theoretical frameworks and empirical studies in sports psychology.

The introduction effectively sets the context by defining sport persistence and highlighting its significance. It nicely leads into the rationale for using Bronfenbrenner's model. However, it mentions the lack of research in "international practice," without specifying what aspects of sport persistence have been overlooked in previous studies and how this study aims to fill those gaps.

The literature review could be further improved by addressing the integration of Bronfenbrenner’s model with the existing literature. Moreover, it could also have more discussion about the critical analysis of existing studies, identifying specific limitations or biases in those studies that the current research addresses.

The study employs a qualitative approach using semi-structured interviews with a large sample size of 133 athletes. I appreciate the authors’ effort to have such a large sample size for qualitative research design. However, the manuscript could elaborate on how it was applied during data analysis, including any coding procedures and how themes were generated.

The discussion and implications are well-addressed. It could be better to discuss the broader societal implications of promoting sport persistence among youth, particularly in relation to physical health and community engagement.

In general, I enjoy reading this manuscript and believe it has the potential to make a valuable contribution to the field. However, I recommend a minor revision to strengthen the articulation of the study's unique contribution, the integration of theoretical frameworks, and the rigor of the qualitative methodology. Good luck. 

Comments on the Quality of English Language

The quality of English language is satisfactory. 

Author Response

Thank you very much for your supportive review. Based on your comments, the following modifications were carried out:

Reviewer 3: The manuscript addresses an under-researched area of sport persistence using a qualitative approach grounded in Bronfenbrenner's socio-ecological model. This is a novel approach, as much of the existing literature focuses on quantitative measures of sporting habits and motivation. By focusing on micro-system level factors, the study provides new insights into the specific roles of family, peers, and coaches in the persistence of young athletes in sports. However, the manuscript lacks a clearer articulation of how its findings differ from or challenge existing theoretical frameworks and empirical studies in sports psychology.

Authors: Thank you very much for your kind feedback. Your suggestions were useful and supported us improving the quality of the paper. Thank you very much for your conscientious evaluation!

Reviewer 3: The introduction effectively sets the context by defining sport persistence and highlighting its significance. It nicely leads into the rationale for using Bronfenbrenner's model. However, it mentions the lack of research in "international practice," without specifying what aspects of sport persistence have been overlooked in previous studies and how this study aims to fill those gaps.

Authors: Thank you very much for your kind feedback. We reflected on the shortcoming of international practice.

Reviewer 3: The literature review could be further improved by addressing the integration of Bronfenbrenner’s model with the existing literature. Moreover, it could also have more discussion about the critical analysis of existing studies, identifying specific limitations or biases in those studies that the current research addresses.

Authors: Thank you very much for your kind notification. The literature review was extended with further relevant resources.

Reviewer 3: The study employs a qualitative approach using semi-structured interviews with a large sample size of 133 athletes. I appreciate the authors’ effort to have such a large sample size for qualitative research design. However, the manuscript could elaborate on how it was applied during data analysis, including any coding procedures and how themes were generated.

Authors: Thank you very much for your kind suggestion. We completed he Methods section with the elaboration of the methodological background.

Reviewer 3: The discussion and implications are well-addressed. It could be better to discuss the broader societal implications of promoting sport persistence among youth, particularly in relation to physical health and community engagement.

Authors: Thank you very much for your kind suggestion. We tried to modify the Discussion and Conclusions following your advice.

Reviewer 3: In general, I enjoy reading this manuscript and believe it has the potential to make a valuable contribution to the field. However, I recommend a minor revision to strengthen the articulation of the study's unique contribution, the integration of theoretical frameworks, and the rigor of the qualitative methodology. Good luck. 

Authors: Thank you very much for your kind feedback and your conscientious work once again.

We hope that we could improve the quality of the manuscript and it can be accepted for publication.

Sincerely,

the Author

Round 2

Reviewer 1 Report

Comments and Suggestions for Authors

First of all, we would like to thank the authors for improving the manuscript. They have been significant improvements. However, there are some details that still need to be worked on. These are:

Work on the title to conform to the journal's standards. You can find it in the form of a comment in the PDF document.

There are two references that are not cited in the document and it is necessary to add them where appropriate and, by the way, check that the whole document is well referenced. These references are number [25] and [33].

It is important to add a subsection that differentiates the conclusions from the limitations and future perspectives and recommendations. This helps the reader to better understand your paper. 

Reviewer 2 Report

Comments and Suggestions for Authors

The authors went above and beyond the requested edits. The authors properly addressed my concerns and were amenable to my feedback. I would like to thank them for their being receptive. It is apparent that they did much work in adding to the richness of the text. I find the authors to be trustworthy and rigorous.